# Effect of current density on the solid electrolyte interphase formation at the lithium|-Li$_6$PS$_5$Cl interface

Sudarshan Narayanan [1,2], Ulderico Ulissi [3], Joshua S. Gibson [1,2], Yvonne A. Chart[1,2], Robert S. Weatherup [1,2] & Mauro Pasta [1,2] ✉

Understanding the chemical composition and morphological evolution of the solid electrolyte interphase (SEI) formed at the interface between the lithium metal electrode and an inorganic solid-state electrolyte is crucial for developing reliable all-solid-state lithium batteries. To better understand the interaction between these cell components, we carry out X-ray photoemission spectroscopy (XPS) measurements during lithium plating on the surface of a Li$_6$PS$_5$Cl solid-state electrolyte pellet using an electron beam. The analyses of the XPS data highlight the role of Li plating current density on the evolution of a uniform and ionically conductive (i.e., Li$_3$P-rich) SEI capable of decreasing the electrode|solid electrolyte interfacial resistance. The XPS findings are validated via electrochemical impedance spectrsocopy measurements of all-solid-state lithium-based cells.

Solid-state battery (SSB) technology incorporating inorganic solid-state electrolytes is fast becoming an attractive option to power electric vehicles (EVs), primarily as it can enable the implementation of lithium metal anodes (theoretical capacity -3860 mAh g$^{-1}$ and 2061 mAh cm$^{-3}$)[1,2]. These can lead to cells with gravimetric and volumetric energies upwards of 400 Wh kg$^{-1}$ and 1000 Wh L$^{-1}$, which are thermally stable and also amenable to fast charging[2–5]. However, Li metal adoption is fraught with issues that hinder its commercialisation. With a strong reduction potential of −3.04 V (vs. standard hydrogen electrode), Li typically reacts with solid electrolytes (SEs) to form kinetically and thermodynamically unstable interphases[6,7]. Combined with other morphological, structural, and chemo-mechanical processes at the Li−SE interface, gradual cell performance degradation and failure follow as a consequence of poor electrode-electrolyte contact, current focusing, mechanical fracture of the SE, inhomogeneous plating/stripping, Li filamentary growth and void formation[6,8–11].

Despite possessing a limited electrochemical stability window[12], the Li$_6$PS$_5$Cl (LPSCl) electrolyte is known to form a kinetically stable solid electrolyte interphase (SEI) when in contact with metallic Li[6,7]. While on one hand, the formation of this SEI is necessary to prevent further SE decomposition, on the other it negatively impacts the

electrochemical performance of the SSB as it significantly increases the interfacial impedance[13–15]. Specifically for LPSCl, the decomposition products are Li$_2$S, LiCl and Li$_x$P, which are poor Li$^+$ ion conductors (bulk ionic conductivity, $\sigma < 10^{-7}$ S cm$^{-1}$ at ambient temperature $\approx 25\,°C$)[16] and directly affect cycling capacities and coulombic efficiencies at anodic potentials[17,18]. Based on thermodynamic considerations, it has been postulated that Li$_6$PS$_5$Cl decomposes via a multi-step mechanism that can broadly be represented by Li$_6$PS$_5$Cl → Li$_2$S + P + LiCl → Li$_3$P, transitioning through partially reduced phosphide species that are typically formed under Li-deficient conditions[19–21]. Interestingly, the fully reduced Li$_3$P has been shown to have an ionic conductivity of $\sigma > 10^{-4}$ S · cm$^{-1}$ at an ambient temperature of 23 °C, while also demonstrating good stability in contact with Li metal[16,22].

Such insights into interfacial degradation phenomena can inform improved engineering of interfaces that are stable even at relatively high current densities ($\geq 0.7$ mA cm$^{-2}$)[22]. Previously, the use of anode interlayers, Li alloys, and high stack pressures have been proposed to tackle the interfacial stability issues in lab-scale cells, albeit at the cost of reduced energy density and/or scalability[23–25]. Recently, a technique employing the use of electrochemical pulses has been reported as a

[1]Department of Materials, University of Oxford, Oxford OX1 3PH, UK. [2]The Faraday Institution Quad One, Harwell Science and Innovation Campus, Didcot OX11 0RA, UK. [3]Nissan Technical Centre Europe, Moulsoe Road, Cranfield Technology Park, Bedford MK43 0DB, UK. ✉e-mail: mauro.pasta@materials.ox.ac.uk

means to decrease interfacial impedance between Li metal and oxide SEs that typically form kinetically and thermodynamically stable interphases, through improved Li−SE contact[26]. Although such interfacial reduction processes are understood to be mainly controlled by reaction kinetics[7,27,28], a systematic investigation of their effects, especially at the Li−SE interface, is still missing. In particular, the evolution of the Li−LPSCl SEI itself as a function of Li metal plating/stripping kinetics is poorly understood[29].

Here, we investigate a current density-mediated evolution of the interphase formed between Li metal and LPSCl sulphide solid electrolyte during electrochemical plating using an in situ X-ray photoemission spectroscopy (XPS) technique. Correlating these results with electrochemical impedance spectroscopy (EIS) measurements and analyses of this process in a "lithium-free" negative electrode solid-state configuration (owing to the negative electrode being devoid of any metallic lithium at the time of assembly) further provides valuable insights into the impact of SEI formation kinetics and composition on the Li−SE interfacial impedance. These findings can support the development of formation protocols to engineer stable interfaces, thereby improving lithium plating and stripping efficiency at high current densities.

The choice of a lithium-free negative electrode configuration is the most suitable and relevant system to investigate the initial formation and subsequent evolution of the SEI. Furthermore, such a system is also attractive from the perspective of increased energy densities, Li-free processing and lower cost[30]. Owing to the dynamic nature of such interfaces, it is imperative that in situ and operando techniques are utilised to adequately characterise the system[11]. The aforementioned in situ XPS method, illustrated schematically in Fig. 1a, has been adapted from one that was previously developed and described in detail by Wood et al. and in other related works[21,31,32]. We also refer to this as a "virtual electrode plating" (VEP) process throughout the rest of this study. This stems from the formation of a "virtual electrode" following exposure of a grounded and Li-backed SE surface to an electron beam. The negatively charged surface thus formed facilitates migration of Li$^+$ ions eventually leading to plating of metallic Li on the SE surface. Furthermore, in this study, the electron beam current (EBC) was adjusted to modulate the electron flux incident at the SE surface, hence tuning the virtual electrode plating current. To observe the electrochemical response due to the evolution of the interface, an effectively identical setup was used in a solid electrolyte cell so as to study the plating of Li at a stainless steel (SS)

current collector (CC) using impedance and potential analyses at various applied current densities (Fig. 1b).

## Results and discussion

### Virtual electrode plating in XPS and SEI evolution

Solid LPSCl pellets (diameter, $\phi_s$ = 5 mm) were first cold-pressed inside an Ar-filled glovebox, to which thin Li metal (~100 μm) and SS foils (25 μm) were attached as CC as shown in Fig. 1a. The assembled pellets for VEP-XPS were then mounted on an XPS sample stage using conductive carbon tape (to ensure a conductive path to the grounded stage) with the exposed LPSCl surface facing the X-ray beam. Detailed experimental procedures can be found in the Methods section. With the EBC set to 30 μA, core-level photoemission spectra for Li $1s$, S $2p$, P $2p$ and Cl $2p$ transitions were acquired at 1-min intervals, over a period of ~18 min. The Ar$^+$ charge neutralisation feature was turned off throughout the experiment. Additionally, spectra pertaining to oxygen and carbon, typically present as trace contaminants in inert gas and vacuum environments, were also collected, since these too provide useful insights into SEI evolution[21,33–35]. Figure 2a (left panel) shows the evolution of the Li $1s$ spectra as the virtual electrode plating progresses. Taking into account the size of the charge-neutralising $e^-$-beam (electron beam, diameter $\phi_{e^-}$ ≈ 5 mm for the BaO electron neutraliser), an equivalent current density ($j_{eq}$) applied to the sample is defined here as $j_{eq} = EBC/a_s$, where $a_s$ is the surface area (here, $a_s = a_{e−beam}$). Accordingly, the amount of Li plated can be estimated from $j_{eq}$, the time of exposure ($t_{exp}$), and expressed in terms of an equivalent areal charge passed, $q_A$, such that $q_A = j_{eq} \cdot t_{exp}$ (μAh cm$^{-2}$).

At 30 μA EBC ($j_{eq}$ ≈ 0.15 mA cm$^{-2}$), the evolution of Li $1s$ spectra in the left panel of Fig. 2a initially depicts a broadening of the peak followed by a gradual shift towards lower binding energies (B.E.). As mentioned earlier, during initial contact with LPSCl, Li metal forms Li$_2$S, Li$_x$P and LiCl, along with other products from reactions with contaminants, such as Li$_2$O and Li$_2$CO$_3$. The peak-broadening observed here can be directly attributed to the formation of these interphasial products[13,34]. Eventually, after the sample has been exposed to $q_A$ ≈ 8.5 μAh cm$^{-2}$, a low B.E. peak appears around ~52.5 eV (the blue area in Fig. 2a) characteristic of metallic Li (Li$^0$) plating at the SE surface. This feature continues to grow in intensity as plating proceeds. Evolution of the Li $1s$ spectra for the same plating process conducted at an EBC of 10 μA ($j_{eq}$ ≈ 0.05 mA cm$^{-2}$) is largely similar to that for $j_{eq}$ = 0.15 mA cm$^{-2}$ (central panel of Fig. 2a), except in the appearance and later growth of

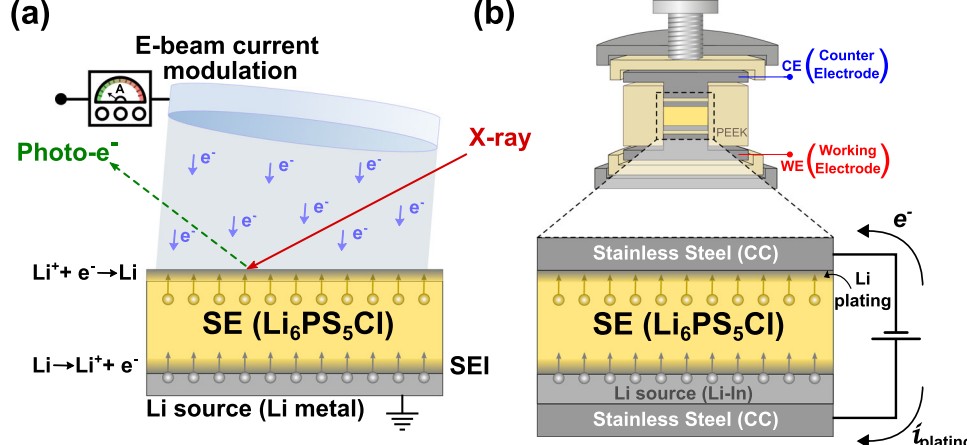

**Fig. 1 | Schematic representation of lithium plating in XPS and a solid electrolyte cell with a lithium-free anode.** Schematics depicting **a** the in situ XPS virtual electrode plating technique where the electron beam current can be modulated while acquiring photoemission spectra from the surface of the solid electrolyte (SE), and **b** the electrochemical setup in a lithium-free anode configuration with Li$_6$PS$_5$Cl SE for studying impedance and electrode potential evolution during plating at the Li-free electrode (stainless steel current collector−CC). The Li-free electrode was used as the counter electrode (CE) and a Li-In layer as the source of Li was used as the working electrode (WE) in this setup. The electrodes and SE were assembled inside a mould made of PEEK (polyether ether ketone).

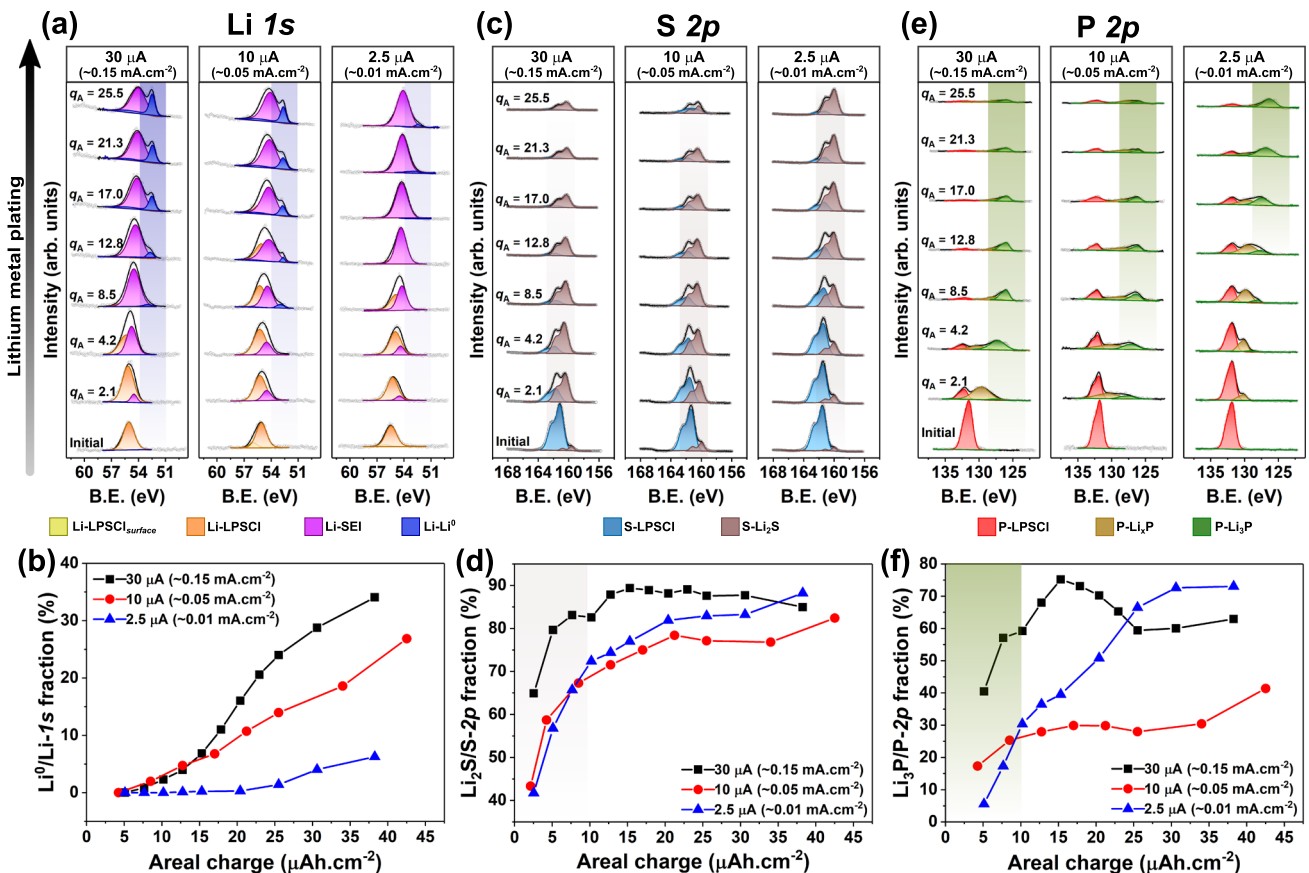

**Fig. 2 | XPS measurements to study SEI evolution during virtual electrode plating at SE surface.** Evolution of core-level XPS spectra during the virtual electrode plating process at the LPSCl surface, at applied EBCs of 30 µA (or ~0.15 mA cm$^{-2}$, left panel), 10 µA (or ~0.05 mA cm$^{-2}$, central panel) and 2.5 µA (or ~0.01 mA cm$^{-2}$, right panel), for **a** Li *1s*, **c** S *2p* and **e** P *2p* transitions, as a function of the charge passed, $q_A$ (µAh cm$^{-2}$). Quantification of XPS spectra plotted over varying amounts of charge passed at different current densities, depicting compositional fraction of **b** metallic Li (Li$^0$) in Li *1s*, **d** Li$_2$S in S *2p*, and **f** Li$_3$P in P *2p*. A larger fraction of Li$^0$ (panel **b**) and Li$_3$P (green area in panel **f**) for small amounts of charge passed at high current densities indicates faster reaction kinetics at the interface resulting in a quicker formation and growth of a metallic Li layer during plating.

the low B.E. Li$^0$ peak. This feature grows in intensity more slowly than when plating at $j_{eq}$ = 0.15 mA cm$^{-2}$, for approximately equivalent charge passed ($q_A$ > 12.8 µAh cm$^{-2}$). This disparity is amplified further on lowering the EBC to 2.5 µA ($j_{eq} \approx 0.01$ mA cm$^{-2}$), where only a negligible Li$^0$ peak is visible even after $q_A$ > 20 µAh cm$^{-2}$ of equivalent charge has been passed (right panel of Fig. 2a). From a qualitative assessment of the Li *1s* spectra, we infer that the formation and growth of a metallic Li layer occurs at $q_A$ < 10 µAh cm$^{-2}$ at high current densities ($j_{eq} \geq 0.1$ mA cm$^{-2}$), while at low current densities, the SEI continues to evolve.

To quantitatively interpret this observation, the Li *1s* spectra were fitted to the following components (using Gaussian and Voigt lineshapes for the pristine LPSCl and SEI layers, and an asymmetric Doniach–Sunjic lineshape for Li metal)—(i) Li at the surface of the as-prepared LPSCl pellet, likely representing products of reaction with surface-adsorbed species, such as carbon, oxygen, CO$_3^{2-}$, HCO$_3^-$, OH$^-$, etc., (ii) Li ions bound within the P-S tetrahedron of the Li$_6$PS$_5$Cl argyrodite structure, labelled as Li–LPSCl, (iii) Li ions as part of the formed interphase, collectively identified as Li-SEI owing to the complexity in deconvoluting individual contributions, and (iv) metallic Li, marked as Li–Li$^0$ [13,34]. Comparing the fraction of metallic Li in the Li *1s* spectra and its evolution as a function of the equivalent charge passed provides quantitative evidence for the accelerated appearance of a metallic Li layer at high current densities (Fig. 2b). In other words, we conclude that Li plates out as a metallic layer faster at high current densities than at low ones.

While the Li *1s* spectra provide insights into Li plating behaviour, the S *2p* and P *2p* spectra shed light on the evolution of the SEI

chemistry. In the case of the S *2p* signal, as indicated in Fig. 2c, with an increasing amount of Li plated, a doublet feature characteristic of Li$_2$S ($2p_{3/2}$ B.E. ~ 160 eV, the brown area in Fig. 2c) develops almost readily. This is a well-reported and studied component of the Li–LPSCl SEI that forms even under Li-deficient conditions [13,21,34]. As the virtual electrode plating process progresses and metallic Li accumulates at the SE surface, the intensity of the Li$_2$S component diminishes owing to the limited depth sensitivity of XPS acquired with an Al Kα source (~13–17 nm for Li-containing compounds, see Supplementary Note 1). On close examination of the composition quantified from spectra measured at the different EBCs as in Fig. 2d, reduction of the LPSCl surface to Li$_2$S occurs at a noticeably faster rate at a high $j_{eq}$ (here, 0.15 mA cm$^{-2}$). Even for low equivalent charge passed ($q_A \approx 5$ µAh cm$^{-2}$), ~90% of sulphur-containing species within the developing SEI comprises Li$_2$S. By comparison, at $j_{eq} \leq 0.05$ mA cm$^{-2}$, only about 70% of the S *2p* spectrum is composed of the reduced sulphide species. This indicates comparatively sluggish reaction kinetics at low current densities driven by various reduction reactions competing for available reactant species (here, plated Li).

These observations bring in to question the actual thickness of the SEI layer thus formed. At a current density of $j_{eq} \approx 0.15$ mA cm$^{-2}$, theoretically, Li could be expected to plate out at an approximate rate of 12 nm/min (or 2.5 µAh cm$^{-2}$ min$^{-1}$). Furthermore, passing $q_A \approx 8.5$ µAh cm$^{-2}$ of equivalent charge would correspond to ~40 nm of Li plated, which is when the signal for metallic Li (at a B.E. of ~ 52.5 eV) appears. The same signal is significantly more intense for $q_A$ > 20 µAh cm$^{-2}$, corresponding to ~ 100 nm of Li plated. With depth sensitivity of XPS

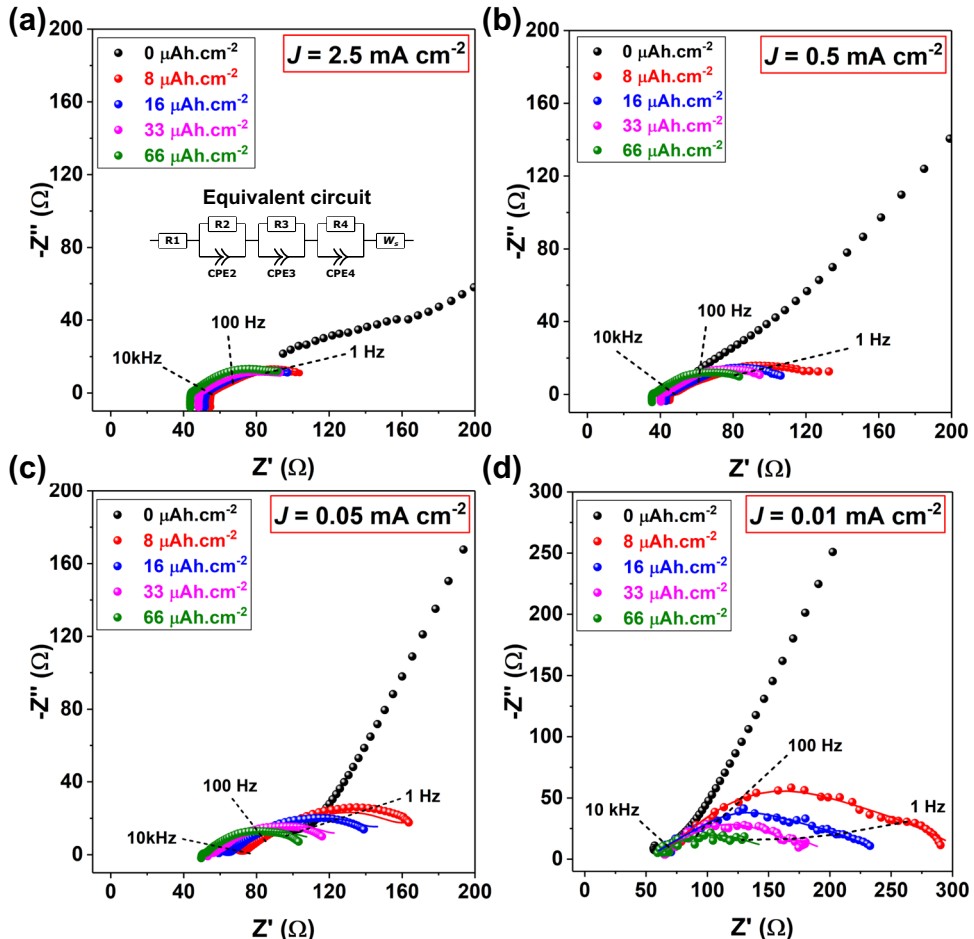

**Fig. 3 | EIS measurements in lithium-free anode cell at current densities 0.01–2.5 mA cm⁻².** Nyquist curves (raw data as symbols, curve fits as lines) depicting evolution of electrochemical impedances for the lithium-free half cell SS|LPSCl|LiIn for plating at **a** $J_{2.5}$ = 2.5 mA cm⁻², **b** $J_{0.5}$ = 0.5 mA cm⁻², **c** $J_{0.05}$ = 0.05 mA cm⁻² and **d** $J_{0.01}$ = 0.01 mA cm⁻², after passing similar equivalent charge. Variation in the low-frequency tail of the impedance curves can be correlated to extent of change in interphasial evolution. The high impedance resulting from poor initial contact with the ion-blocking electrode at the Li-less anode side is evident from EIS curves at $q_A$ = 0 μAh cm⁻² (black circles)[50]. The equivalent circuit used to fit the obtained EIS data is schematically depicted as an inset in panel a and corresponding values are reported in Supplementary Table 2, followed by a brief discussion presented in Supplementary Note 5[40,41].

for Li and Li-containing compounds at these kinetic energies being of the order of 15–20 nm (see Supplementary Note 1), and with S *2p*, P *2p* and Cl *2p* spectral intensities attenuating significantly for $q_A$ > 35 μAh cm⁻², the SEI could be estimated to be ~150–200 nm thick. Recent work by Otto et al. on the investigation of the Li–LPSCl interface using time-of-flight secondary ion mass spectroscopy (ToF-SIMS) and atomic force microscopy (AFM) estimate the SEI thickness to be 250 ± 25 nm (see Supplementary Table 1)[36], which is ten times greater than previously suggested by Wenzel et al.[13]. In light of such evidence reported in literature, it can be speculated that the SEI formed during plating of Li metal with LPSCl is of the order of 100–200 nm at least. Additionally, the relatively large inelastic mean-free path of photoelectrons at high kinetic energies (low binding energies) extends the XPS information depth for Li metal and Li-containing compounds. This further complicates any reasonable estimation of SEI thicknesses using XPS alone.

Meanwhile, phosphorous in the P-S tetrahedron has been reported to follow a less direct route towards reduction to Li₃P upon reaction with the plating Li by forming multiple partially reduced species, as discussed earlier, that are often collectively labelled as Li$_x$P[19–21]. Comparing evolution in the P *2p* spectra reveals a swift reduction to a low B.E. doublet feature ($2p_{3/2}$ ≈ 126 eV) at the highest $j_{eq}$, for $q_A$ < 5 μAh cm⁻² of equivalent charge passed (Fig. 2e, left panel, green area), representative of the fully reduced Li₃P. In contrast, for $j_{eq}$ ≤ 0.05 mA cm⁻² at similar amounts of charge passed, the initial SEI formed more prominently comprises a broad spectral feature (126 eV < B.E. $2p_{3/2}$ < 131 eV) characteristic of the partially reduced Li$_x$P. Continued plating eventually results in the formation of the fully reduced Li₃P at low current densities as well, accompanied by a large decrease in the overall P *2p* spectral intensity, suggesting accumulation of plated metallic Li. These observations imply the formation of a Li₃P-rich SEI in the early stages of its formation for Li plated at high current densities even at low equivalent charge passed. The fraction of Li₃P quantified as a function of the equivalent charge passed, particularly for $q_A$ < 10 μAh cm⁻² (Fig. 2f) provides evidence supporting this conclusion. In Fig. 2c and 2e, it can also be seen that XPS signal pertaining to pristine LPSCl components are more rapidly attenuated in intensity at high $j_{eq}$ for similar equivalent charge passed (e.g., $q_A$ = 12.8 μAh cm⁻²). With XPS spectra acquired over an area of 500 μm × 500 μm, representing a large sampling surface, it can be inferred that the SEI layer formed is comparatively more uniform and homogeneous for Li plated at high current densities. We note that the suppression of P *2p* photoemission signal with increased Li plating (Fig. 2e), does result in a lower signal-to-noise ratio. Accordingly, subsequent errors in component fitting limit the validity of such a comparison of fitted components, especially of S and P, to low equivalent charge regimes ($q_A$ < 10 μAh cm⁻², highlighted regions in Fig. 2d and 2f).

The partially reduced Li$_x$P species observed are not transient or metastable either. Supplementary Fig. 1 shows that the corresponding

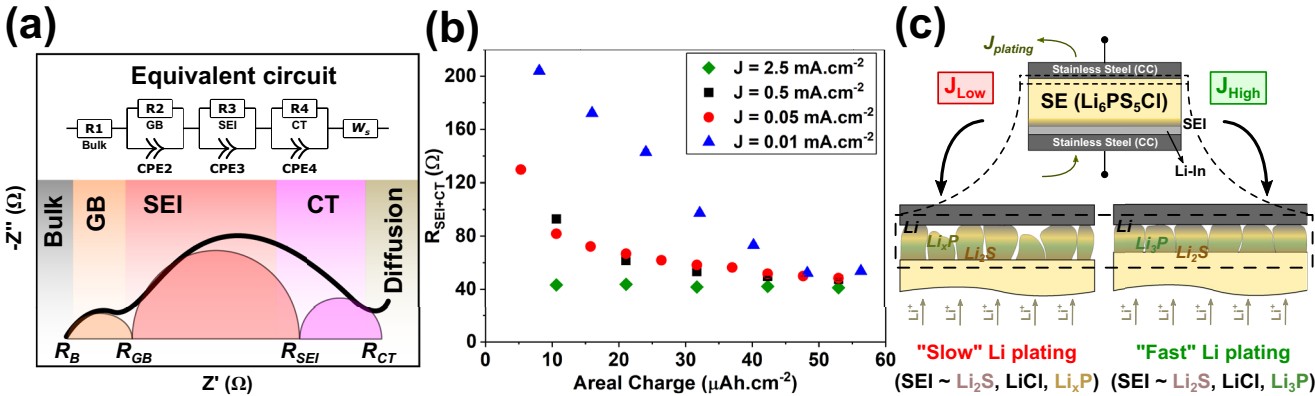

**Fig. 4 | Interpretation of EIS spectra. a** Equivalent circuit used for fitting the EIS data along with a schematic describing the impedance contributions from the bulk ($R_B$), grain boundaries ($R_{GB}$), SEI ($R_{SEI}$) and charge transfer ($R_{CT}$) processes[40,41]. **b** Variation in interfacial resistance, $R_{int}$ ($R_{SEI}$ + $R_{CT}$), from fitted EIS data as a function of the amount of charge passed during Li plating at current densities, $J_{2.5}$ = 2.5 mA cm$^{-2}$, $J_{0.5}$ = 0.5 mA cm$^{-2}$, $J_{0.05}$ = 0.05 mA cm$^{-2}$ and $J_{0.01}$ = 0.01 mA cm$^{-2}$ [40,41]. Parameters obtained from EIS fits along with error estimates between raw and fitted data are tabulated in Supplementary Table 2. **c** Schematic representation of the likely mechanism of SEI formation and Li plating as a function of applied current density, at $J_{2.5}$ ($J_{High}$) and $J_{0.01}$ ($J_{Low}$).

P *2p* feature remains stable and largely unperturbed even after a 24-h period when left under vacuum, inside the XPS. Also, the presence of the partially reduced phosphide buried under the surface of a ~20 nm Li layer thermally evaporated on the LPSCl surface, from spectra obtained using a tunable-energy synchrotron X-ray source, is further testament to the reactive stability of the species in the absence of an electrochemical plating/stripping event (Supplementary Fig. 2a–c). Although with the passage of adequate charge the SEI is rendered self-limiting[6], the evolution of the core-level XPS spectra for Li, S, and P indicate significant variations in the SEI composition itself for the virtual electrode plating conducted at $j_{eq}$ ranging from high (0.15 mA cm$^{-2}$) to low (0.01 mA cm$^{-2}$). Plausible reasons to explain these observations could be twofold: (a) the inherent kinetics and energetics associated with the reaction between Li metal and sulphide SEs, and (b) the thermodynamics of nucleation and growth of Li metal at the SE surface governed by electrochemical overpotential and surface energy considerations[7]. While thermodynamic forces are no doubt crucial in understanding Li–SE interfacial evolution, we hypothesise that reaction kinetics play a dominant role in determining the SEI evolution phenomena observed in this work. This has important ramifications in the context of overall interfacial impedance of the cell which will be discussed in later sections.

Similar results were obtained on probing the SEI evolution at the Li–SE interface by in situ sputtering of Li[13], where the Li sputter rate was controlled by varying the Ar$^+$ ion acceleration voltage (see Supplementary Fig. 3a–d and associated discussion in Supplementary Note 2). However, since surface damage by the more energetic Li atoms produced by sputtering has been shown to affect interfacial reactions[35], direct correlation of observed SEI evolution with varying flux of incident Li atoms is not feasible in such a setup. Another by-product of the reaction between Li and LPSCl is LiCl, which cannot be easily discerned using XPS owing to the binding energies of chlorine in both LPSCl and LiCl being nearly identical, as shown in Supplementary Fig. 4a[13,34,37]. Though an inevitable contaminant[34,35], the evolution of O *1s* spectra as a function of progression in the virtual electrode plating process also confirms the disparity in Li reaction kinetics, with the formation of Li$_2$O species (at a B.E. of ~528.5 eV) proceeding at a faster rate for less equivalent charge passed at high EBCs (see Supplementary Fig. 4b). In this regard, we direct the readers to Supplementary Note 3 and Supplementary Fig. 5a–d for a brief discussion on the interaction of Li with contamination on the SE surface as well as with the ambient environment in the XPS chamber during the virtual electrode plating process.

### Electrochemical plating in cells with the lithium-free anode

To evaluate the impact of current density-dependent interphasial evolution on the electrochemical properties of the system, we assembled and tested a SS|SE|LiIn cell with a "lithium-free" anode configuration, as shown in Fig. 1b. Here, a LiIn alloy (Li$_{0.25}$In$_{0.75}$) was used as a source of Li owing to its reasonable interfacial stability with sulphide SEs under practically applicable stack pressures (≤10 MPa)[24,38] and subjected to current densities as high as 2 mA cm$^{-2}$ (see ref. 39), thus allowing for isolation of changes occurring at the SS|LPSCl side[23]. Closely replicating the virtual electrode plating experiment in the XPS, the evolution of cell impedance was studied during plating of Li at the SS ion-blocking electrode on the anode side by applying different current densities (identified here as $J_{0.01}$ = 0.01 mA cm$^{-2}$, $J_{0.05}$ = 0.05 mA cm$^{-2}$, $J_{0.5}$ = 0.5 mA cm$^{-2}$ and $J_{2.5}$ = 2.5 mA cm$^{-2}$). The cells were allowed to stabilise over a period of 24–48 h (see Supplementary Fig. 6a–e) prior to plating and their open circuit voltages were monitored as a function of charge passed (Supplementary Fig. 6f). A solid-state symmetric cell (SS|LPSCl|SS) was also setup for comparison (see Supplementary Fig. 7a, b) demonstrating the ion-blocking nature of the SS electrodes. As in the case of XPS spectra discussed earlier, the Nyquist plots also show marked differences in their evolution for equivalent amounts of charge passed, depending on the current density applied (Fig. 3a–d). The high-frequency intercept with the real axis, which would correspond to the bulk impedance of the cell, can be expected to evolve as lithium metal plates onto the SS current collector. This can be attributed to a progressively improving contact between the SS foil and the SE. Schlenker et al. have previously reported such evolution as lithium metal plates onto an electrode[40]. Whereas at relatively low current densities ($J$ = 0.01–0.05 mA cm$^{-2}$ in this study, Fig. 3c, d), the low-frequency tail of the impedance curves shows a gradual decrease in Re(Z) (real component of impedance, or Z') as Li plating proceeds, at higher current densities ($J ≥ 0.5$ mA cm$^{-2}$ in this study, Fig. 3a, b), for an equivalent amount of Li plated at the SS current collector, the same low-frequency component of Re(Z) converges to a stable value faster, with this effect being most prominent at the highest tested current density, $J_{2.5}$.

### Current density-dependent evolution of interfacial impedance

In order to understand the underlying cause of this trend, the Nyquist plots obtained were fitted to the equivalent circuit schematically depicted in Fig. 4a[41]. This circuit consists of a single resistor in series with three parallel circuits of a resistor and a constant-phase element (CPE) along with an additional Warburg diffusion ($W_S$) component.

Schlenker et al. attribute the inclusion of the latter to an impedance at low frequencies arising from a lithium vacancy diffusion gradient generated most likely at the interface between the LiIn and LPSCl[40]. In this equivalent circuit, one of the parallel circuits ($R_2||CPE_2$) in combination with the individual resistor $R_1$ can be assigned to the bulk and grain boundary resistance, which typically exhibits low effective capacitances (correlating to the term $Q_{CPE} \sim 10^{-6} F.s^{a-1}$, with $a$ as the constant phase, as estimated from the CPE component of the EIS spectral data fit). The other circuit element pairs ($R_3||CPE_3$ and $R_4||CPE_4$) exhibiting relatively higher effective capacitances ($Q_{CPE} \sim 10^{-4} \text{-} 10^{-2} F.s^{a-1}$) can be understood to represent the LPSCl-Li interface and charge transfer (CT) processes, respectively, as plating begins with the formation of an SEI[40–42]. The variations in impedance can thus be attributed largely to the interface, which comprises the SEI and CT components that are represented by the low-frequency segment of the spectra[40–42]. The interfacial impedance ($R_{int}$) can then be approximated as $R_{int} = R_{SEI} + R_{CT}$[41], where $R_{CT}$ is related to the intrinsic kinetics of the system[28,43,44]. Indeed, a plot of combined resistances from SEI and CT contributions (Fig. 4b) suggests that at low current densities (in this study, $J_{0.5}$ and $J_{0.05}$), the interfacial resistance asymptotically reaches a minimum. In stark contrast, for Li plating conducted at a significantly higher current density ($J_{2.5}$), the interface attains the same minimum resistance almost as soon as Li begins plating. It is noteworthy that the differences in impedance evolution are most prominent over ~50 µAh cm$^{-2}$ of charge passed, beyond which the effect diminishes as expected, while plating proceeds to form a metallic Li layer in both cases. Thus, a rapid drop in interfacial resistance, combined with the appearance of fully reduced reaction products ($Li_3P$ in particular) and the presence of a greater fraction of metallic Li in XPS analyses, all observed within ~10 µAh cm$^{-2}$ of charge passed during initial stages of plating, strongly suggest the formation of a more uniform and homogeneous SEI layer as well, for Li plated at high current densities (Fig. 4c).

Recalling an earlier discussion on overpotential-dictated Li nucleation and growth behaviour during plating[7,43,45], overpotentials due to current density variations in this study were also measured (Supplementary Fig. 6f) but their contribution to the SEI evolution observed here are beyond the scope of this work and will be the subject of a future study. Meanwhile, the evolution of electrochemical impedance observed here in conjunction with that of SEI chemistry measured using the XPS validates the assumption of a kinetically mediated reaction process governing the Li–SE interface. This understanding is also in good agreement with previous reports that have shown, through finite element simulations, that the presence of a $Li_3P$ membrane layer on the electrode surface results in a more homogeneous electric field distribution, enabling more uniform Li plating/stripping[22]. The validity of this hypothesis was experimentally verified by testing the utilisation efficiencies of the anodes formed in an identical cell setup with a lithium-free anode. The experiment involved plating Li at two current densities ($J_{0.05}$ and $J_{2.5}$) and then stripping both electrodes at the same current density ($J_{Stripping} = 0.25$ mA cm$^{-2}$). Supplementary Fig. 8a, b demonstrates that almost twice the amount of Li was successfully stripped out from the cell plated at a higher current density. An ex situ examination of scanning electron micrographs (SEM) of the SS foil surface peeled off from the LPSCl surface, after passing $q_A \approx 30$ µAh cm$^{-2}$ of charge at $J_{2.5}$ and $J_{0.05}$ (Supplementary Fig. 9a, b, respectively) reveals a pattern of Li islands covered with SE material (Supplementary Fig. 9c, d for Li plating at $J_{2.5}$ and $J_{0.05}$, respectively) adhered to them similar to that described in ref. 40. Here, for the sample plated at $J_{2.5}$ the islands appear relatively smaller and more uniformly distributed spatially, supporting the argument of homogeneity in Li plating at higher current densities (see Supplementary Note 4).

In conclusion, we report here a current density-mediated evolution of the interphase formed from contact of Li metal with an $Li_6PS_5Cl$ SE introduced using an in situ virtual electrode electrochemical plating method during XPS measurements, that suggests a significant role of reaction kinetics in these processes. At high current densities, in the initial stages of plating, this interphase was found to be rich in $Li_3P$, a fully reduced decomposition product of $Li_6PS_5Cl$. Evidenced by the appearance of metallic Li plated at the SE surface combined with the complete suppression of spectroscopic signal from the pristine LPSCl surface at relatively low equivalent charge passed, the interphase at high current densities is understood to be effectively more uniform as well. Analysis of interfacial impedances via EIS measurements in conjunction with XPS spectra lend further credence to such an assertion. Moreover, the SEI formed at higher current densities, consisting of the $Li^+$ ion conducting $Li_3P$, is inferred to be more homogeneous. We believe this understanding can be leveraged to suitably engineer electrode-electrolyte interfaces and develop charge-discharge protocols, particularly in lithium-free SSBs.

## Methods

### Materials

$Li_6PS_5Cl$ (LPSCl) argyrodite sulphide solid electrolyte powder was purchased from Ampcera™ (D50 ~1 µm) through MSE Supplies LLC., and used as received. Li foil (30-µm thick, 99.9%) procured from Honjo Metal Co. Ltd. through KISCO GmbH was used as received to prepare samples for XPS and electrochemical characterisation. SS foil that was used as an ion-blocking electrode and current collector, was purchased from Advent Research Materials Ltd. (SS316 of composition Fe/Cr18/Ni10/Mo3, low carbon, temper annealed, 25-µm thick). The $Li_{0.25}In_{0.75}$ (LiIn) alloy was produced by first heating In metal powder (Merck Life Sciences, 99.99% purity) to 600 °C (~100 °C above the liquidus temperature for this composition[23]) in a crucible inside a furnace placed within an Ar-filled glovebox (MTI-KSL-1200X). The molten In was removed from the furnace, and stoichiometric amount of Li (25 at.%) purchased from Merck Life Sciences (750-µm thick, 99.9% purity) was added and stirred into the melt. The molten mixture was returned to the furnace for a further 2 h period at 600 °C. Following this, the melt from the crucible was directly poured onto a 0.5-mm thick (30 cm × 30 cm) stainless steel plate to allow rapid solidification. The obtained melt-processed mixture was sealed inside aluminium-laminated pouches under vacuum (using MSK-115-III sealer from MTI Corp), and calendered into foils of thickness ~50 µm at ambient temperatures of 25 ± 0.5 °C (using MTI cold rolling press, MSK-2150-PD, inside an Ar-filled glovebox), for subsequent use.

The Ar-filled gloveboxes used in the study were maintained at ~4 mbar positive pressure containing < 0.1 ppm of $H_2O$ and < 0.1 ppm of $O_2$. All materials were handled in the same high-purity environment.

### Sample preparation

For VEP-XPS measurements, LPSCl pellets of diameter 5 mm (~25 mg to obtain ~700–750-µm thick pellets) were cold-sintered using a hydraulic press (YLJ-40TA-PE from MTI Corp.), at 500 MPa for 5 min. Li foil from Honjo Metal Co. Ltd. was pressed onto one side of the LPSCl pellet mechanically by applying a pressure of ~50–70 MPa for ~30 s, after punching out a 5-mm diameter disc (Rapid-core 5.0 mm sampling tool, EM Sciences). In addition, a 5 mm diameter disc of SS was pressed against the Li foil by hand (~10–20 MPa). The assembled stacks were then mounted onto the XPS sample holder using conductive carbon tape.

Sample stacks for electrochemical measurements were prepared by pressing solid electrolyte powder (~100 mg to obtain ~700–750-µm thick pellets) within a 10-mm diameter polyether ether ketone (PEEK) mould, along with an SS disc of the same size to be used as the plating electrode, at 500 MPa for 5 min using a hydraulic press. Further, a disc of of LiIn foil of diameter 10 mm was punched out (10-mm round punch, EK tools) and then used

as the counter electrode, pressed with an SS current collector by application of ~80–100 MPa of pressure. The assembly of the electrochemical cell was completed by placing SS plungers in contact with both current collectors, and inserted into a custom cell setup described in the main text (Fig. 1b, pictorially represented in Supplementary Fig. 10[46]. An external pressure of 10 MPa was applied to this cell stack assembly by means of a screw tightened to an appropriate torque using a torque wrench.

## X-ray photoemission spectroscopy and analysis

X-ray photoemission spectroscopy (XPS) was conducted using a PHI Versaprobe III XPS system generating focused, monochromatic Al $K\alpha$ X-rays at 1486.6 eV, under ultrahigh vacuum (UHV) conditions with the main chamber maintained at pressures between ~$10^{-7}$–$10^{-6}$ Pa. The X-ray monochromator was operated at a power of 25 W and an electron beam voltage of 15 kV. The instrument is equipped with dual beam charge neutralisation capabilities—a low-energy BaO electron source (beam diameter ~ 5mm) and a low-energy $Ar^+$ ion source. Samples were transferred from a glovebox into the XPS chamber using a vacuum transfer vessel to avoid contamination and any ambient exposures. Survey scans were acquired at pass energies of 224 eV, whereas for core-level spectra, a lower pass energy of 55 eV was used. For VEP-XPS experiments, all scans for the survey and core-level spectra were acquired with both neutralisers switched off. Acquired spectra were then fitted to Gaussian–Lorentzian and Voigt lineshapes (or Doniach–Sunjic lineshapes for metallic Li components, where asymmetry was found to be significant), after application of a Shirley background, using CasaXPS software[47].

All spectra were charge referenced to adventitious C at 284.8 eV through acquired C 1s spectra, and validated with that of Cl 2p spectra for Cl $2p_{3/2}$ = 198.5 eV, since Cl 2p signal remains unchanged throughout the Li plating process[13,21]. The validity of the choice of Cl 2p spectra for charge correction and peak calibration follows from Supplementary Fig. 11, where the maximum difference in binding energy between any two fitted Cl $2p_{3/2}$ peaks was ~0.16 eV even when the highest EBC of 30 μA (or 0.15 mA cm$^{-2}$) was applied (Supplementary Fig. 11a). Further, Supplementary Fig. 11b shows that the corresponding binding energy shifts after charge correction to Cl $2p_{3/2}$ at 198.5 eV (B.E.$_{raw}$ − B.E.$_{corrected}$) were within $\Delta$B.E. ≈ 3.0 ± 0.1 eV for all three exposure currents used. The evolution of the raw XPS spectra acquired for Li 1s, S 2p and P 2p transitions following exposure to an electron beam current of 30 μA (Supplementary Fig. 11c) are essentially identical to those depicted in Fig. 2a, c, e, reiterating the appropriateness of this method for processing spectra. Following this, fitted regions were quantified and used for estimating relative fractions of components therein.

## In situ virtual electrode plating (VEP)

"Virtual electrode plating" (VEP) was conducted by alternating XPS spectra acquisitions with exposure of the sample stack surface to the low-energy electron beam. The electron dose was further modulated by adjusting the beam current between 2.5 μA and 30 μA, with the latter being the highest current for the safe and stable operation of the neutraliser filament. The accuracy of the applied current during the virtual electrode plating process was verified by measuring the same through a Faraday cup mounted on the sample stage connected to a picoammeter assembly. The measured current was within ± 1 μA of the set currents. During the VEP-XPS experiment, $Ar^+$ charge neutralisation was turned off.

## Electrochemical measurements in cells with the lithium-free anode

Electrochemical impedance spectroscopy (EIS), open circuit voltage (OCV) and galvanostatic plating/stripping measurements were performed on cell stacks that were assembled within a PEEK mould into a

custom cell setup. The cell stacks comprise of LPSCl powder (~100 μg) pressed at 500 MPa for 5 min along with a SS foil on one side (negative electrode) using a hydraulic press. This was followed by the attaching of LiIn foil along with SS foil current collector on the positive electrode side, by applying a pressure of ~80–100 MPa to ensure good contact. The cells were connected in a two-probe configuration to a Gamry Instruments Interface-1000 potentiostat for this purpose. Potentiostatic EIS (PEIS) measurements were conducted in the frequency range of 1 MHz to 0.1 Hz (15 points per decade) with a sinusoidal voltage perturbation of amplitude 10 mV. Initially, PEIS data were acquired between 2 h OCV measurements over a 24 h period to ascertain cell stability. Following this, PEIS spectra were measured immediately after passing indicated amounts of equivalent Li charge (~8 μAh cm$^{-2}$ intervals) at each current density. All EIS data were collected while the assembled cells were stationed within an Ar-filled glovebox, maintained at 25 ± 1 °C, with ~4 mbar positive pressure ensuring < 0.1 ppm of $H_2O$ and < 0.1 ppm of $O_2$. For each condition tested, at least two cells were assembled to ensure reproducibility.

Fitting of EIS data was done using the Z fit functionality in EC-lab® software v11.33 using an equivalent circuit schematically depicted in Fig. 4a. The root mean square error (RMSE) between the raw and fitted data for -Im(Z) (or -Z") vs. Re(Z) (or Z'), for $J_{2.5}$, $J_{0.5}$, $J_{0.05}$, and $J_{0.01}$ have been summarised in Supplementary Table 2, along with a discussion in Supplementary Note 5.

## Synchrotron XPS measurements and analysis

High-energy XPS data were acquired at the I09 beamline in Diamond Light Source synchrotron facility (Didcot, UK), using hard X-rays (hard X-ray photoemission spectroscopy - HAXPES) with photon energies of 2.2 keV and 6.6 keV, as well as soft X-rays (soft X-ray photoemission spectroscopy - SOXPES, with photon energy 450 eV) with kinetic energy of core-level photoelectrons being 315 eV. High-energy X-ray sources can provide non-destructive depth-resolved information about buried interfaces as well. For example, typical inelastic mean-free paths through Li metal for the different photon energies range from ~1.1 nm for SOXPES, ~6.0–7.5 nm for 2.2 keV-HAXPES and ~18.5–19.8 nm for 6.6 keV-HAXPES[48]. Lithium thin film (~20 nm thick) was thermally evaporated by resistively heating to ~ 700 °C Li chips (16 mm dia., 1 mm thick, 99.9% purity, MTI Corp.) placed inside a molybdenum boat (length 32mm, thickness 0.05mm, Agar scientific). The thin film was deposited onto a ~750 μm LPSCl pellet (prepared by pressing ~75 mg of powder at 500 MPa for 5 min using a hydraulic press) using a custom-built ultrahigh vacuum (UHV) chamber attached to the beamline end station[35]. The X-ray beam intensity was defocused to prevent beam damage, with long-term irradiation observed to cause no change in photoelectron peak shapes.

## In situ XPS Li sputtering and analysis

In situ Li sputtering was conducted using a deposition sample holder supplied by the XPS manufacturer, similar to the setup described by Wenzel et al. previously[13,33]. Li foil (750-μm thick, 99.9% purity) from Sigma Aldrich was cut to size (~5mm x 7mm) and mounted onto the sample holder wall using conductive adhesive carbon tape (0.15-mm thickness, Agar Scientific). The foils were also manually scraped using a generic nylon-bristled toothbrush to remove any surface contamination and passivation layers comprising oxides, carbonates, bicarbonates, and other organic/inorganic residues from handling, prior to transferring into the XPS chamber. XPS analysis of even the scraped Li foil surface was found to contain high amounts of $Li_2O$ (~5.3 at.%) and $Li_2CO_3$ (~26 at.%) in addition to some adventitious carbon (~21 at.%), as depicted in Supplementary Fig. 12a–c. The carbonaceous species were, however etched away after just 5 min of exposure to an $Ar^+$ ion beam accelerated to 4 kV (Supplementary Fig. 12a–c), suggesting that such a surface is unlikely to effect any further contamination during subsequent etch cycles.

The in situ sputtering of Li was conducted using an $Ar^+$ ion gun rastered over a 3 mm × 3 mm area, and the accelerating voltage of the ions was adjusted between 2 kV and 4 kV. Estimates for the sputtered film thicknesses were obtained by sputtering Li onto thin Cu foils (~12.5 μm, 99.9% purity, Advent Research Materials) that in turn were pressed onto the LPSCl pellets, with regions of the pellet surface exposed. Using an adaptation of the Strohmeier equation for determining coating layer thicknesses using XPS spectral peak areas (i.e., intensities) with Beer Lambert's law for correlating intensity attenuation of an electromagnetic wave to the thickness of the layer of propagation, an estimate for deposition rate is arrived with the relation $t_{Li} = -\lambda_{Cu}^{Li} \ln(I_{Cu}^{Li}/I_{Cu})$. Here, $t_{Li}$ is the thickness of deposited Li film, $\lambda_{Cu}^{Li}$ is the inelastic mean-free path of Li at the kinetic energy (K.E.) for core-level photoemission from Cu, $I_{Cu}^{Li}$ is the intensity of core-level spectra for Cu with Li layer and $I_{Cu}$ the intensity of core-level spectra for uncoated Cu substrate[49]. Using this method, the deposition rates for Li at 4 kV and 2 kV were estimated to be ~1.4 Å/min and ~0.7 Å/min, respectively.

### Ex situ scanning electron microscopy (SEM) measurements

SEM measurements were conducted using a Thermo-Fisher Helios G4-CXe Plasma FIB (PFIB) instrument with energy-dispersive X-ray spectroscopy functionality. SS foils with electrochemically plated Li from the aforementioned sample stacks were carefully peeled off from the LPSCl surface and mounted on conductive adhesive carbon tape (0.15-mm thickness, EM Sciences) placed on dedicated SEM stubs. The stubs were then mounted on a transportable SEM stage inside an Ar-filled glovebox, before inserting into and sealing within a Gatan iLoad sample transfer vessel. The vessel was transported over to the microscopy room in a custom suitcase designed to protect the vessel from impact. The stage was then transferred from the vessel and into the SEM chamber via a load-lock setup engineered for the iLoad transfer vessel.

## Data availability

The authors declare that all data supporting the findings of this study are included within the paper and its Supplementary Information. Source data are available from the corresponding author (M.P.) upon reasonable request.

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

## Acknowledgements

The authors acknowledge the support of The Faraday Institution (M.P., S.N. and Y.A.C. via grant numbers FIRG026, FIRG020 and FIRG011) as well as the Henry Royce Institute (R.S.W. and J.S.G. through UK Engineering and Physical Science Research Council grant EP/R010145/1) for capital equipment. This project has received funding from the European Research Council (ERC) under the European Union's Horizon 2020 research and innovation programme (J.S.G. and R.S.W. via EXISTAR, grant agreement no. 950598). M.P., S.N. and U.U. acknowledge support from Innovate UK (project number 98841), Nissan Motor Co. Ltd., Japan, Nissan Technical Center Europe, UK. J.S.G. is thankful to Diamond Light Source, DIDCOT, UK, for awarding beamtime (SI 25807-2, SI 25807-3) at their synchrotron facility. S.N. is grateful to Dr. Ben Schmidt and the team at Physical Electronics (PHI) for their support with the VEP-XPS technique.

## Author contributions

S.N. and M.P. conceived the idea and designed the experiments. S.N. performed the VEP-XPS experiments, electrochemical measurements, and their analyses. U.U. provided frequent inputs and helpful discussion on the conduct and analysis of VEP-XPS measurements. J.S.G. conducted synchrotron XPS measurements and analysis. Y.A.C. collected and analysed ex situ SEM micrographs. R.S.W. and J.S.G. provided expertise relating to XPS spectral analysis and data interpretation. S.N. wrote the manuscript with input from all authors. M.P. supervised the design of the project and provided frequent input on the interpretation of all results.

## Competing interests

The authors declare no competing interests.
