## [Peer Review File · Nature Communications]

nature portfolio

Peer Review FileREVIEWER COMMENTS

Reviewer #1 (Remarks to the Author):

I'd like to congratulate the authors for a really well performed experimental analysis of not only the general stability, but also to some extent the kinetic evolution of LiPSCI in contact with lithium. The comparison of different approaches and experimental conditions is quite convincing. It had several critical remarks after the first reading, but most of them were actually answered in the paper after reading carefully again.

Id do have a few points the authors may consider:

1. The current density of 0.15 mA/cm^2 should result to $\pm 12 \text{ nm}$ of Lithium plating per minute, if I calculated correctly (this is something the authors should definitely calculate), thus if successful the XPS signal of the LPSCI should vanish after 1 minute of deposition. However, the signal of Li-metal starts appearing after plating more than 40 nm of Lithium, if the currents are correct. What happens to all the other Lithium? That's far more Lithium atoms than all other elements seen by XPS, while the XPS signals show only slight reduction of those elements. Can the authors think of a possible explanation for this discrepancy? Was the measured stage current in line with the applied $30 \mu\text{A}$?

2. The authors do not give quantification information of the surface composition before and after deposition. However, this may be quite relevant for the reader, since the expected behavior when plating lithium in an UHV atmosphere would be a gradual reaction with the residual background atmosphere, which the authors do comment on. In addition, the initial surface composition may of course influence the degradation process. Without numbers it is however very difficult for the reader to judge the implication of those reaction products and surface contaminants. This information can be put into the supplementary information, but it should be given somewhere.

3. In the 2nd paragraph of the introduction ref 11 and 12 seem to be mixed up, they are used correctly in the rest of the manuscript.

4. Page 5 middle paragraph: I was irritated by the high amount of Li_2S supposedly in the SEI (70-90%), but I believe the authors mean "70-90 % of the sulfur signal of the SEI can be attributed to Li_2S ", at least according to fig 2e. This should be corrected.

Thanks for the paper and all the best, I already learned quite a bit from your work,
Joachim Sann

Reviewer #2 (Remarks to the Author):

This work address a novel methodology that have been reported recently in some previous study by Wood and collaborators. It has been also reported not with such accuracy in few conferences such ECASIA this year and JNSP (using Auger spectroscopy). The concept of using e-gun to activate electrochemically the lithium plating is already reported. However, the current work is rigorous in term of methodology, which bring some potential to the work.

All data in this work are well treated and accurately measured.

Therefore, many issues in this methodology are not clear even in the previous cited works. Before going to detailed comments. There are few questions and remarks.

1- Why the pressure in the XPS chamber is quite low 10^{-7} to 10^{-6} Pa?

2- The term *operando* here is not appropriate, when the XPS are conducted; the term *in situ* is more adapted since there is no external electrochemical sollicitation.

3- The argerodite, used as solid electrolyte is known to be insulator, therefore when we use charge compensation electron gun to conduct *in situ* lithium oxidation, the calculated dose does not take in account the part of electrons that combine at the surface to compensate the residual charge induced by photoemission process. The calculated dose must be considered taking in account the annealed electron at the surface and also the lateral electron diffusion.

4- This question, bring new open question; all spectra were calibrated to $\text{Cl}2p$ of LPS. How about the relationship between surface potential evolution during "so-called lithium plating" and LPS reduction and core-levels peak shifts. Is there any trend in the spectra recorded in figure 2, before any peak calibration?

5- In figure 2, it seems that the thickness of lithium is less than 5 nm even at $30 \mu\text{A}$, equivalent to $0.15\text{mA}/\text{cm}^2$ (according to the authors). Considering the real electrochemical $\text{Li}/\text{electrolyte}/\text{electrode}$, with such current density, we are able to move in theory (plate) ~ 750 nm of lithium. Since $1 \text{mA}/\text{cm}^2$ allowed to move $\sim 5\mu\text{m}$ of lithium. First question, why such low amount of lithium. Is it due to the underestimated current density on the surface or because the probed phenomenon is due to mainly surface reduction of LPS under e-gun exposure. The

6- This brings to experimental check-up. Did the authors perform the same experiment on simple LPS with bottom contact with lithium?

7- The reduction of LPS is fast even in the beginning even without detection of lithium metal, how can this be explained.

Additional remarks are linked to the correlation between *in-situ* deposition followed by HAXPES and EIS measurement.

In the case of *in-situ* lithium deposition, did the authors check the surface of lithium that used to deposit lithium? Usually the lithium has surface passivation constituted of thick Li_2O and Li_2CO_3 layer. The signature of $\text{Li}1s$ deposited lithium indicates the deposition of Li_2O ($\text{O}1s$ peak). We do not observe in this case in this case the same reactivity. The LPS related peak remained almost stable with certain reduction, already mentioned before. Argerodite, known to be unstable to lithium. In this case the SEI is mainly constituted of Li_2O .

Regarding the EIS measurement, at the high frequency, we observe that the contact are not good in almost all case, this could indicate instability of the interfaces during measurement. Did the authors perform similar experiment using symmetrical $\text{Stainless-steel}/\text{LPSCI}/\text{Stainless-steel}$ or $\text{In}/\text{LPSCI}/\text{In}$ cell?

The tails in all EIS spectra keep evolving; signature of continuous reactivity at the interface during lithium plating. For example in the EIS spectra at $8\mu\text{Ah}/\text{cm}^2$ about 50 nm of lithium could be plated. In the EIS spectra, does not look like. Where that lithium goes? What is the critical deposited lithium thickness to allow stable SEI between Li/LPSCI .

Regarding all these comments, this paper has great potential for more specialized journal.

Response to reviewer comments

We would like to thank the reviewers for their constructive and insightful comments. Our responses to questions and clarifications sought have been presented as below. Changes implemented to the original document are highlighted in **yellow** for the direct quote to the revised manuscript.

Reviewer #1:

I'd like to congratulate the authors for a really well performed experimental analysis of not only the general stability, but also to some extent the kinetic evolution of LiPSCI in contact with lithium. The comparison of different approaches and experimental conditions is quite convincing. It had several critical remarks after the first reading, but most of them were actually answered in the paper after reading carefully again.

We thank the reviewer for their comments on the manuscript.

I do have a few points the authors may consider:

1. The current density of 0.15 mA/cm² should result to ±12nm of Lithium plating per minute, if I calculated correctly (this is something the authors should definitely calculate), thus if successful the XPS signal of the LPSCI should vanish after 1 minute of deposition. However, the signal of Li-metal starts appearing after plating more than 40 nm of Lithium, if the currents are correct. What happens to all the other Lithium? That's far more Lithium atoms than all other elements seen by XPS, while the XPS signals show only a slight reduction of those elements. Can the authors think of a possible explanation for this discrepancy? Was the measured stage current in line with the applied 30 µA?

The reviewer is correct in pointing out that, at a current density of 0.15 mA/cm², theoretically 12 nm/min of Li (2.5 µAh/cm²/min) should be electrochemically plated. Furthermore, passing q_A~8.5 µAh/cm² of equivalent charge would indeed correspond to ~40 nm of Li plated, which is when the signal for metallic Li (at a B.E. of ~52.5 eV) appears. The same signal is significantly more intense for q_A>20 µAh/cm² corresponding to ~100 nm of Li plated. The accuracy of the applied current was verified by measuring the same through a Faraday cup mounted on the sample stage connected to a picoammeter assembly. The measured current was within ±1 µA of the set currents.

The presence of XPS signal from other elements in spite of the large amount of Li metal plated can be explained using the following arguments:

1. In recent work, Otto et al. investigated the Li-LPSCI interface (among other solid electrolytes) in-situ using ToF-SIMS and AFM [Otto2022]. Here, they report that the thickness of the Li₂S-rich SEI formed with LPSCI is ~250±25 nm, for Li plated using an electron flood gun operated at 10 µA (~0.1 mA/cm²) over 1h, inside the SIMS instrument. This thickness estimate is approximately 10 times greater than previous estimates of such SEI thicknesses [Wenzel2018]. In related work cited in the main manuscript, Wood et al. also observe the appearance of metallic Li signal after ~2h of exposure to the electron flood gun (~0.17 mA/cm²), when conducting a similar plating experiment on LPS solid electrolyte.
2. The binding energies of characteristic transitions of elements probed in this study, such as Cl 2p, S 2p, P 2p and Li 1s, are <200 eV. This corresponds to kinetic energies >1200 eV for the emitted photoelectrons. This results in relatively long inelastic mean-free paths (λ_{imfp}) for photoelectrons in Li and Li-containing compounds [Tanuma2011], of the order of ~45-55 Å. Further, the effective XPS probing depth (3λ_{imfp}) at these

energies can be ~13-17 nm [Greczynski2020]. Thus, even with a large amount of Li plated, the XPS spectra constitute signals from depths up to 15-20 nm under the surface.

In light of such evidence reported in literature, it can be speculated that the SEI formed during plating of Li metal with LPSCI is of the order of 100-200 nm at least. Additionally, the relatively large inelastic mean-free path of photoelectrons at high kinetic energies (low binding energies) extends the XPS information depth for Li metal and Li-containing compounds. This further complicates any reasonable estimation of SEI thicknesses using XPS alone.

The discussion above has been included in the main text and reads thus (Page 5) -

“These observations bring to question the actual thickness of the SEI layer thus formed. At a current density of $j_{eq} \approx 0.15 \text{ mA}\cdot\text{cm}^{-2}$, theoretically, Li could be expected to plate out at an approximate rate of 12 nm/min (or $2.5 \mu\text{Ah}\cdot\text{cm}^{-2}\text{min}^{-1}$). Furthermore, passing $q_A \approx 8.5 \mu\text{Ah}\cdot\text{cm}^{-2}$ of equivalent charge would correspond to ~40 nm of Li plated, which is when the signal for metallic Li (at a B.E. of ~52.5 eV) appears. The same signal is significantly more intense for $q_A > 20 \mu\text{Ah}\cdot\text{cm}^{-2}$ corresponding to ~100 nm of Li plated. With depth sensitivity of XPS for Li and Li-containing compounds at these kinetic energies being of the order of 15-20 nm (see Supplementary Note 1), and with S 2p, P 2p and Cl 2p spectral intensities attenuating significantly for $q_A > 35 \mu\text{Ah}\cdot\text{cm}^{-2}$, the SEI could be estimated to be approximately 150-200 nm thick. Recent work by Otto et al. on investigation of the Li-LPSCI interface using time-of-flight secondary ion mass spectroscopy (ToF-SIMS) and atomic force microscopy (AFM) estimate the SEI thickness to be $250 \pm 25 \text{ nm}$ (see Supplementary Note 2) [Otto2022], which is 10 times greater than previously suggested by Wenzel et al.¹² In light of such evidence reported in literature, it can be speculated that the SEI formed during plating of Li metal with LPSCI is of the order of 100-200 nm at least. Additionally, the relatively large inelastic mean-free path of photoelectrons at high kinetic energies (low binding energies) extends the XPS information depth for Li metal and Li-containing compounds. This further complicates any reasonable estimation of SEI thicknesses using XPS alone.”

“Supplementary Note 1: The binding energies of characteristic transitions of elements probed in this study, such as Cl 2p, S 2p, P 2p and Li 1s, are <200 eV. This corresponds to kinetic energies >1200 eV for the emitted photoelectrons. This results in relatively long inelastic mean-free paths (λ_{imfp}) for photoelectrons in Li and Li-containing compounds [Tanuma2011], of the order of ~45-55 Å. Further, the effective XPS probing depth ($3\lambda_{imfp}$) at these energies can be ~13-17 nm [Greczynski2020]. Thus, even with a large amount of Li plated, the XPS spectra constitute signals from depths up to 15-20 nm under the surface.”

“Supplementary Note 2: Prior work on characterisation of Li-SE interfaces indicate a lack of general consensus on the thickness of SEI formed for comparable SE materials. The table below summarises some prominent findings from these reports.

Sl. No.	Solid Electrolyte	Characterisation Method	SEI Thickness	Ref.	Remarks
1	Li ₆ PS ₅ Cl	XPS	~23 nm	2	In situ sputtered Li
2	Li ₆ PS ₅ Cl	ToF-SIMS + AFM	~250 ± 25nm	7	Electron-beam induced plating
3	Li ₆ PS ₅ Cl	XPS	~151 nm	3	Ex-situ XPS on peeled Li

4	LiPON	Cryo-EM	~80 nm	8	Cryogenic lift-out of Li-LiPON interface
---	-------	---------	--------	---	--

2. The authors do not give quantification information of the surface composition before and after deposition. However, this may be quite relevant for the reader, since the expected behavior when plating lithium in an UHV atmosphere would be a gradual reaction with the residual background atmosphere, which the authors do comment on. In addition, the initial surface composition may of course influence the degradation process. Without numbers it is however very difficult for the reader to judge the implication of those reaction products and surface contaminants. This information can be put into the supplementary information, but it should be given somewhere.

It is true that within an UHV atmosphere, the plating lithium would be expected to gradually react with the existing environment. It is to be noted here that most of the initial contamination present on the surface of the sample can be attributed to the glovebox environment or even the inert transfer vessel used to transfer between the glovebox and the XPS. This has been extensively studied in a separate work by us.[Gibson2022] In the same study it was also shown that after lithium is introduced (by evaporation, or by sputtering, or by a virtual electrode plating as done here), there is little difference in the reaction products formed.

Here too, as suggested by the reviewer, we include the surface composition before and after plating of Li ($\sim 25.5 \mu\text{Ah}\cdot\text{cm}^{-2}$) in Supplementary Figure 6a. It can be seen that despite slightly varying initial compositions, especially in the case of O and C, the compositions for species present in the SE (Li, S, P and Cl) are either similar or follow a trend that is suggestive of current density-dependent reactivities with Li. Supplementary Figures S6b-d provide a more detailed idea of the evolution of composition with increasing plated Li. Even in the case where initial C contamination was relatively large (beam current = 10 μA), trends in composition evolution for Li, S, P and Cl are similar throughout - distinct differences in final amounts of Li emerge along with corresponding differences in S concentration (attributable to Li bound as Li_2S).

This discussion has also been included in the SI as Supplementary Note and reads thus -

“Supplementary Note 3: Despite being an UHV environment, the XPS chamber ambient contains trace oxygen, moisture and other carbon contaminants. During the operando virtual electrode plating process, lithium would be expected to gradually react with the existing environment. It is to be noted here that most of the initial contamination present on the surface of the sample can be attributed to the glovebox environment or even the inert transfer vessel used to transfer between the glovebox and the XPS.[Wood2018a,Gibson2022] In a separate study, Gibson et al. demonstrate that there are but minor differences in reaction products formed on introduction of Li to the SE surface (by evaporation, or by sputtering, or by a virtual electrode plating setup).[Gibson2022]”

Supplementary Figure 6a shows a comparison of the SE surface composition before and after plating of Li ($\sim 25.5 \mu\text{Ah}\cdot\text{cm}^{-2}$). It can be seen that despite slightly varying initial compositions, especially in the case of O and C, the compositions for species present in the SE (Li, S, P and Cl) are either similar or follow a trend that is suggestive of current density-dependent reactivities with Li. Supplementary Figures S6b-d provide more insight into the evolution of composition with increasing plated Li. Even in the case where initial C contamination was relatively large (beam current = 10 μA), trends in composition evolution for Li, S, P and Cl are similar throughout - distinct differences in final amounts of Li emerge along with corresponding differences in S concentration (attributable to Li bound as Li_2S).”

Supplementary Figure 6: (a) Comparison of the SE surface composition before and after plating of Li ($\sim 25.5 \mu\text{Ah.cm}^{-2}$) estimated by quantifying regions from XPS spectra for O 1s, C 1s, Cl 2p, S 2p, P 2p and Li 1s transitions. Evolution of SE surface composition as a function of charge passed ($\mu\text{Ah.cm}^{-2}$) during the virtual electrode plating process for applied EBC of (b) 30 μA , (c) 10 μA and (d) 2.5 μA .

3. In the 2nd paragraph of the introduction ref 11 and 12 seem to be mixed up, they are used correctly in the rest of the manuscript.

We acknowledge the reviewer's note on the positioning of the references - 11 (Banerjee et al. Chem Rev., 2020) and 12 (Wenzel et al. Solid State Ionics, 2018). However, we believe that these are appropriately used in this particular instance. Banerjee et al. in their review on interfaces and interphases in ASSBs discuss the various mechanisms of interaction between inorganic solid electrolytes and Li metal, wherein they suggest that $\text{Li}_6\text{PS}_5\text{Cl}$ forms a kinetically stable solid electrolyte interphase (SEI) when in contact with metallic Li. Meanwhile, Wenzel et al. in their seminal work on interfacial reactivity and interphasial growth for argyrodite-type SEs, report approximately threefold increase in SEI resistance over a 12 hour period of contact between $\text{Li}_6\text{PS}_5\text{Cl}$ and Li. They also indicate that R_{SEI} more or less stabilises after that period, while continuing to grow at a much slower rate. Further, Yu et al. and Kato et al. allude to this aspect negatively impacting electrochemical performance of SSBs fabricated with $\text{Li}_6\text{PS}_5\text{Cl}$ as SE.

4. Page 5 middle paragraph: I was irritated by the high amount of Li_2S supposedly in the SEI (70-90%), but I believe the authors mean "70-90 % of the sulfur signal of the SEI can be attributed to Li_2S ", at least according to fig 2e. This should be corrected.

We thank the reviewer for pointing out this discrepancy. Indeed, 70-90% of the composition of the sulphur species as estimated from analysis of the XPS spectra can be attributed to Li₂S. We have accordingly modified the text in the manuscript.

Even for low equivalent charge passed ($q_A \approx 5 \mu\text{Ah}\cdot\text{cm}^{-2}$), ~90% of sulphur-containing species within the developing SEI comprises Li₂S. By comparison, at $j_{eq} \leq 0.05 \text{ mA}\cdot\text{cm}^{-2}$, only about 70% of the S 2p spectrum is composed of the reduced sulphide species.

Reviewer #2:

This work address a novel methodology that have been reported recently in some previous study by Wood and collaborators. It has been also reported not with such accuracy in few conferences such ECASIA this year and JNSP (using Auger spectroscopy). The concept of using e-gun to activate electrochemically the lithium plating is already reported. However, the current work is rigorous in term of methodology, which bring some potential to the work. All data in this work are well treated and accurately measured. Therefore, many issues in this methodology are not clear even in the previous cited works.

We thank the reviewer for their review of this work and for the valuable feedback provided.

Before going to detailed comments. There are few questions and remarks.

1. Why the pressure in the XPS chamber is quite low 10⁻⁷ to 10⁻⁶ Pa?

The PHI Versaprobe III XPS instrument used in this work is connected to a Pfeiffer HiPACE 700 turbomolecular pump capable of pumping speeds of 685 l/s (N₂), thus enabling an ultra-high vacuum environment inside the main chamber. Pressures of 10⁻⁷ - 10⁻⁶ Pa (or 10⁻⁹ - 10⁻⁸ mbar) can thus be easily attained and is quite common for most such XPS instruments. We direct the reviewer to related works where similar equipment has been used that can also achieve similar vacuum levels.

- a. PHI Versaprobe 5600: K. N. Wood and G. Teeter, ACS Appl. Energy Mater. 2018, 1, 9, 4493–4504
- b. PHI Versaprobe II: Wenzel et al. Solid State Ionics 318 (2018) 102–112

2. The term *operando* here is not appropriate, when the XPS are conducted; the term *in situ* is more adapted since there is no external electrochemical solicitation.

We acknowledge the reviewer's comment on the appropriateness of the terminologies used in this manuscript. Drawing a distinction between the terms *in situ* and *operando* is no doubt tricky, as we note in prior work (Narayanan et al. Current Opinion in Solid State and Materials Science 26 (2022) 100978). This is because, as described in the Results and Discussion section, exposure of a grounded and Li-backed LPSCI SE pellet to an electron beam leads to the formation of a negatively charged surface (owing to the insulating nature of the SE). This in turn acts as a "virtual electrode" facilitating migration of Li⁺ ions from the Li source on the other end of the SE to the exposed SE surface, eventually resulting in plating of Li metal. Thus, the external stimulus (here, the e-beam) does effect an electrochemical reaction.

Whereas, the term *in situ* (or, "at the site of") can be used even for a reaction or process occurring at a surface without any electrical/chemical stimulus, as in the case of the study of evolution of interphase formed on sputtering Li metal onto a SE surface (reported here in the Supporting Information section, and by others such as Wenzel et al. Solid State Ionics 318 (2018) 102–112 and Connell et al. Chem. Mater. 2020, 32, 23, 10207–10215).

Thus, neither term is technically accurate in this context. Notably, it has been acknowledged that especially in the batteries community, there has been no agreement or consensus on the

most appropriate language to describe such characterisation methods.[Peterson2021] Hence, we have de-emphasised the term “operando” and replaced it with “in situ virtual electrode plating” in the manuscript to provide more clarity on the method used in this work.

3. The argerodite, used as solid electrolyte is known to be insulator, therefore when we use charge compensation electron gun to conduct in situ lithium oxidation, the calculated dose does not take in account the part of electrons that combine at the surface to compensate the residual charge induced by photoemission process. The calculated dose must be considered taking in account the annealed electron at the surface and also the lateral electron diffusion.

The reviewer raises a valid point in stating that it would be necessary to account for various electronic processes occurring at the surface when considering the X-ray photoemission flux and compensation of residual charge induced, using an electron gun. However, owing to the complexity in deconvoluting instrumental factors from measured intensities of the XPS spectra, estimating the true dose of electrons at the surface is extremely challenging and beyond the scope of this study.

Nevertheless, in control experiments where an LPSCI pellet grounded to the XPS stage without a Li foil backing on the other side was exposed to similar electron beam currents ($\sim 10 \mu\text{A}$) no sample charging was observed, and correspondingly, no changes in the shape of the spectra were noticed. This can be explained by the fact that, although categorised as an insulator, the electronic conductivity of $\text{Li}_6\text{PS}_5\text{Cl}$ has been estimated to be $\sim 10^{-9}$ - 10^{-8} S/cm which is at least 2-3 orders of magnitude higher than that of typical insulators and also SEs like LiPON.[Gorai2021] Further, from computational studies, $\text{Li}_6\text{PS}_5\text{Cl}$ is a native n-type material with excess free electrons, where the free electron concentration has been shown to range between 10^7 - 10^8 cm^{-3} , [Gorai2021] only ~ 100 times less than that of intrinsic Si, a semiconductor.[Misiakos1993] Based on this reasoning, it can be inferred that electron loss from photoemission can be easily compensated by the SE's free electrons in this case. Also, the electron dose from the electron-beam will primarily facilitate ionic migration and not interfere with the photoemission process.

4. This question, bring new open question; all spectra where calibrated to Cl2p of LPS. How about the relationship between surface potential evolution during “so-called lithium plating” and LPS reduction and core-levels peak shifts. Is there any trend in the spectra recorded in figure 2, before any peak calibration?

We have considered the influence of core-level peak shifts induced by surface potential changes during the Li-plating process. The choice of Cl 2p for charge correction and peak calibration was supported by the absence of any significant change in the evolution of the corresponding spectra for all three electron beam exposure currents used.

As can be seen in the figure that we are including in the Supplementary Information file, Supplementary Figure 7), the maximum difference noticed between any two Cl 2p spectra was ~ 0.16 eV when the maximum electron beam current of $30 \mu\text{A}$ (or $0.15 \text{ mA}\cdot\text{cm}^{-2}$) was applied (Supplementary Figure 7a). Further, Supplementary Figure 7b shows that the corresponding binding energy shifts after charge correction to Cl $2p_{3/2}$ at 198.5 eV ($\text{B.E}_{\text{raw}} - \text{B.E}_{\text{corrected}}$) were within $\Delta\text{B.E.} \sim 3.0 \pm 0.1$ eV for all three exposure currents used. The evolution of the raw XPS spectra acquired for Li 1s, S 2p and P 2p transitions following exposure to an electron beam current of $30 \mu\text{A}$ (Supplementary Figure 7c) are essentially identical to those depicted in Figures 2a-c in the main text.

Based on these observations we believe that charge correction using the Cl 2p transition is valid and that it does not lead to any notable differences in the spectral evolution for other core-level transitions (such as Li 1s, S 2p or P 2p). A clarification regarding this has been included in the manuscript.

“The validity of the choice of Cl 2p spectra for charge correction and peak calibration follows from Supplementary Figure 7, where the maximum difference between any two Cl 2p spectra was ~ 0.16 eV when the highest EBC of $30\ \mu\text{A}$ (or $0.15\ \text{mAcm}^{-2}$) was applied (Supplementary Figure 7a). Further, Supplementary Figure 7b shows that the corresponding binding energy shifts after charge correction to Cl $2p_{3/2}$ at $198.5\ \text{eV}$ ($\text{B.E.}_{\text{raw}} - \text{B.E.}_{\text{corrected}}$) were within $\Delta\text{B.E.} \sim 3.0 \pm 0.1$ eV for all three exposure currents used. The evolution of the raw XPS spectra acquired for Li 1s, S 2p and P 2p transitions following exposure to an electron beam current of $30\ \mu\text{A}$ (Supplementary Figure 7c) are essentially identical to those depicted in Figures 2a-c, reiterating the appropriateness of this method for processing spectra.”

Supplementary Figure 7: (a) Evolution of XPS spectra pertaining to the Cl 2p transitions during the virtual electrode plating process at the LPSCI surface, at an applied EBC of $30\ \mu\text{A}$, with and without charge-correction to the Cl $2p_{3/2}$ peak at $198.5\ \text{eV}$. (b) The computed binding energy shifts after charge correction to Cl $2p_{3/2}$ at $198.5\ \text{eV}$ ($\text{B.E.}_{\text{raw}} - \text{B.E.}_{\text{corrected}}$) for all three EBCs used for virtual electrode plating ($2.5\ \mu\text{A}$, $10\ \mu\text{A}$, and $30\ \mu\text{A}$) as a function of charge passed, q_A ($\mu\text{Ah.cm}^{-2}$). (c) Evolution of XPS spectra pertaining to other core level transitions, Li 1s, S 2p and P 2p, for an EBC of $30\ \mu\text{A}$, as acquired and without post-processing or charge-correction.

5. In figure 2, it seems that the thickness of lithium is less than $5\ \text{nm}$ even at $30\ \mu\text{A}$, equivalent to $0.15\ \text{mA/cm}$ (according to the authors). Considering the real electrochemical Li/electrolyte/electrode, with such current density, we are able to move in theory (plate) $\sim 750\ \text{nm}$ of lithium. Since $1\ \text{mA/cm}^2$ allowed to move $\sim 5\ \mu\text{m}$ of lithium. First question, why such low amount of lithium. Is it due to the underestimated current density on the surface or because the probed phenomenon is due to mainly surface reduction of LPS under e-gun exposure.

As explained in response to a similar question earlier, it is true that at an equivalent current density of $0.15\ \text{mA/cm}^2$, ideally, approximately $12\ \text{nm/min}$ of Li could be estimated to have been plated after passing $q_A \sim 2.5\ \mu\text{Ah/cm}^2/\text{min}$ of equivalent charge. We would like to clarify

that in this study, as shown in Figure 2, the overall duration of exposure to the electron beam at the highest current tested (30 μA) was for ~ 15 mins ($q_A \sim 37.5 \mu\text{Ah}/\text{cm}^2$ of equivalent charge), which would correspond to ~ 180 nm of Li plated. During the virtual electrode plating process, we start to notice the signal for metallic Li (at a B.E. of ~ 52.5 eV) appear after passing $q_A \sim 8.5 \mu\text{Ah}/\text{cm}^2$ of equivalent charge, or ~ 40 nm of Li plated.

While the typical information depth for XPS spectra is ~ 5 - 10 nm, it is difficult to make an accurate estimation of the depth of the surface layer. In fact, information depth itself can vary widely with the inelastic mean-free path (λ_{imfp}) for photoelectrons at a particular kinetic energy. In the case of Li and Li-containing compounds, [Tanuma2011] this can be of the order of ~ 13 - 17 nm ($\sim 3\lambda_{\text{imfp}}$ at kinetic energies >1200 eV). [Greczynski2020] In other words, even after plating ~ 40 nm of Li, information from the XPS spectra acquired can easily be from depths up to 15 - 20 nm underneath the surface being probed.

As to the current density on the surface of the SE, we verified the value of the applied current on the sample by separately measuring the same through a Faraday cup mounted on the sample stage connected to a picoammeter. The measured current was within $\pm 1 \mu\text{A}$ of the set currents, based on which we believe that the estimated current densities are reasonably accurate.

We clarify these details in the manuscript as well.

“These observations bring to question the actual thickness of the SEI layer thus formed. At a current density of $j_{\text{eq}} \approx 0.15 \text{ mA}\cdot\text{cm}^{-2}$, ideally, Li could be expected to plate out at an approximate rate of $12 \text{ nm}/\text{min}$ (or $2.5 \mu\text{Ah}\cdot\text{cm}^{-2}\text{min}^{-1}$). Furthermore, passing $q_A \approx 8.5 \mu\text{Ah}\cdot\text{cm}^{-2}$ of equivalent charge would correspond to ~ 40 nm of Li plated, which is when the signal for metallic Li (at a B.E. of ~ 52.5 eV) appears. The same signal is significantly more intense for $q_A > 20 \mu\text{Ah}\cdot\text{cm}^{-2}$ corresponding to ~ 100 nm of Li plated. With depth sensitivity of XPS for Li and Li-containing compounds at these kinetic energies being of the order of 15 - 20 nm (see Supplementary Note 1), and with S 2p, P 2p and Cl 2p spectral intensities subsiding significantly for $q_A > 35 \mu\text{Ah}\cdot\text{cm}^{-2}$, the SEI could be estimated to be approximately 150 - 200 nm thick. Recent work by Otto et al. on investigation of the Li-LPSCI interface using time-of-flight secondary ion mass spectroscopy (ToF-SIMS) and atomic force microscopy (AFM) estimate the SEI thickness to be $\approx 250 \pm 25$ nm (see Supplementary Note 2) [Otto2022], which is 10 times greater than previously suggested by Wenzel et al.¹² In light of such evidence reported in literature, it can be speculated that the SEI formed during plating of Li metal with LPSCI is of the order of 100 - 200 nm at least. Additionally, the relatively large inelastic mean-free path of photoelectrons at high kinetic energies (low binding energies) extends the XPS information depth for Li metal and Li-containing compounds. This further complicates any reasonable estimation of SEI thicknesses using XPS alone.”

“Supplementary Note 1: The binding energies of characteristic transitions of elements probed in this study, such as Cl 2p, S 2p, P 2p and Li 1s, are <200 eV. This corresponds to kinetic energies >1200 eV for the emitted photoelectrons. This results in relatively long inelastic mean-free paths (λ_{imfp}) for photoelectrons in Li and Li-containing compounds [Tanuma2011], of the order of ~ 45 - 55 Å. Further, the effective XPS probing depth ($3\lambda_{\text{imfp}}$) at these energies can be ~ 13 - 17 nm [Greczynski2020]. Thus, even with a large amount of Li plated, the XPS spectra constitute signals from depths up to 15 - 20 nm under the surface.”

“Supplementary Note 2: Prior work on characterisation of Li-SE interfaces indicate a lack of general consensus on the thickness of SEI formed for comparable SE materials. The table below summarises some prominent findings from these reports.”

Sl. No.	Solid Electrolyte	Characterisation Method	SEI Thickness	Ref.	Remarks
1	Li ₆ PS ₅ Cl	XPS	~23 nm	S4	In situ sputtered Li
2	Li ₆ PS ₅ Cl	ToF-SIMS + AFM	~250 ± 25nm	S?	Electron-beam induced plating
3	Li ₆ PS ₅ Cl	XPS	~151 nm	S7	Ex-situ XPS on peeled Li
4	LiPON	Cryo-EM	~80 nm	S?	Cryogenic lift-out of Li-LiPON interface

6. This bring to experimental check-up. Did the authors performed the same experiment on simple LPS with bottom contact with lithium?

In this study we have chosen Li₆PS₅Cl (LPSCI) as the SE for studying the effect of current density on the SEI chemistry, owing to its promise as a high ionic conductivity SE that is easy to process and is scalable. We have not extended the study to other SEs such as LPS, since Wood et al. (Nat Commun 9, 2490 (2018)) have previously conducted a detailed study on the SEI evolution using the virtual electrode plating process on LPS as the SE. However, while they note that varying the EBC had little effect on the SEI chemistry, on close scrutiny of data presented in Supplementary Figure 9 of their Supplementary Information file, we contend that there is evidence for distinct evolution profiles for the same amount of charge passed. A copy of the graphic (from Wood et al.'s work) is presented below with regions showing differences in compositional ratios within the SEI highlighted for comparison.

Supplementary Figure 9. Comparison of SEI evolution under different 'virtual electrode' electron kinetic energies. Current density for the high electron kinetic energy case is about 10 times higher than for the low-energy case. For equivalent amounts of total charge, Q (~600 mC), nearly identical SEI phases are present in both cases.

7. The reduction of LPS is fast even in the beginning even without detection of lithium metal, how can this be explained.

Assuming that the reviewer is referring to LPSCI which is the system of study here, we discuss in some detail the interaction of Li metal with LPSCI in the manuscript. From various prior reports on this system, LPSCI is known to react almost instantaneously with Li metal to form a kinetically stable SEI layer. [Wenzel2018, Tan2019, Banerjee2020, Schwietert2020] Specifically, Banerjee et al. inform that for LPSCI, reduction processes can occur at voltages below 1.7V versus Li+/Li, and that under phase equilibrium the reduction products would

contain Li_2S , Li_3P and LiCl . Owing to the high thermodynamic stability of these binary compounds with Li metal, such a phase equilibrium can be achieved even at 0V. Using AIMD simulations, Cheng et al. (ACS Energy Lett. 2017, 2, 6, 1454–1459) suggest that the quick decomposition of LPSCI in contact with Li metal can be attributed to the weak bonding between P and S. However, since the SEI formed thus is kinetically stable, as demonstrated by Wenzel et al., additional charge passed leads to an eventual plating out of lithium metal.

8. Additional remarks are linked to the correlation between in-situ deposition followed by HAXPES and EIS measurement. In the case of in-situ lithium deposition, did the authors check the surface of lithium that used to deposit lithium? Usually the lithium has surface passivation constituted of thick Li_2O and Li_2CO_3 layer. The signature of $\text{Li}1s$ deposited lithium indicates the deposition of Li_2O ($\text{O}1s$ peak). We do not observe in this case the same reactivity. The LiPS related peak remained almost stable with certain reduction, already mentioned before. Argerodite, known to be instable to lithium. In this case the SEI is mainly constituted of Li_2O .

In this study, the *in situ* Li sputtering experiment within the XPS was conducted using lithium foil whose surface was scraped and cleaned immediately prior to transferring into the XPS chamber. This can typically remove most organic and inorganic contamination. However, owing to the high reactivity of lithium metal, as the reviewer has rightly pointed out, a passivation layer comprising Li_2O and Li_2CO_3 can easily form even under UHV conditions. We have conducted XPS on bare Li foil that had been scraped and cleaned, before and after etching with an Ar^+ ion beam. While the surface does appear to contain small amounts of Li_2O and Li_2CO_3 in addition to adventitious carbon, the carbonaceous species are etched away after just 5 mins of etching with Ar^+ ions accelerated to 4kV.

Even in the case of evaporation of Li prior to synchrotron XPS measurements, the custom built UHV chamber attached to the I09 beamline end station at Diamond Light Source was equipped with a shutter to prevent evaporation of contaminated species on the lithium source during the initial stages of deposition.

For these reasons, we believe that neither deposition process from such Li sources would result in any additional contamination on the SE surface. A note to this effect has been included in the Methods section under "In situ XPS Li sputtering and analysis" -

Li foil (750 μm thick) from Sigma Aldrich were cut to size and mounted on to the sample holder wall using carbon tape. The foils were also scraped using a brush to remove any surface contamination and passivation layers comprising oxides, carbonates, bicarbonates, and other organic/inorganic residues from handling, prior to transferring into the XPS chamber. XPS analysis of the scraped Li foil surface was found to contain only small amounts of Li_2O and Li_2CO_3 in addition to some adventitious carbon (Supplementary Figure 4a-c). The carbonaceous species were however etched away after just 5 mins of exposure to an Ar^+ ion beam accelerated to 4kV, suggesting that such a surface is unlikely to effect any further contamination during subsequent etch cycles.

Supplementary Figure 4: XPS spectra for scraped and cleaned Li foil used for in situ sputtering in the pristine and etched states (4kV Ar+ ion beam) for (a) Li 1s, (b) O 1s, and (c) C 1s transitions.

In response to the reviewer's comment on presence of Li_2O from O 1s spectra, we agree that Li_2O is indeed a component of the SEI, as has previously been reported by Wood et al. and others. However, correlating Figure 2a and Supplementary Figure 5b, it can be argued that signal for Li_2O becomes prominent only when lithium metal starts plating out (for e.g., at $q_A \sim 8.5 \mu\text{Ah}/\text{cm}^2$ of equivalent charge passed at $\text{EBC} = 30 \mu\text{A}$). This suggests that the increase in Li_2O signal intensity is more likely from the reaction of plated metallic Li on the surface with ambient oxygen in the chamber. This also implies that SEI evolution initially starts with formation of Li_2S , Li_3P and LiCl , which constitute the bulk of the SEI layer.

9. Regarding the EIS measurement, at the high frequency, we observe that the contact are not good in almost all case, this could indicate instability of the interfaces during measurement. Did the authors performed similar experiment using symmetrical Stainless-steel/LPSCI/ Stainless-steel or In/LPSCI/In cell? The tails in all EIS spectra keep evolving; signature of continuous reactivity at the interface during lithium plating. For example in the EIS spectra at $8 \mu\text{Ah}/\text{cm}^2$ about 50 nm of lithium could be plated. In the EIS spectra, does not looks like. Where that lithium goes? What is the critical deposited lithium thickness to allows stable SEI between Li/LPSCI.

In this work, we studied the evolution of impedance during lithium plating onto a stainless steel (SS) current collector, in an asymmetric "lithium-less" cell configuration. The contact between the solid current collector and the solid electrolyte will be poor. Nevertheless, we have updated Supplementary Figure 8a after having repeated EIS stability over a 24 hour period for instances where these were still found to be questionable. As can be seen in the updated plots, the initial EIS is stable and similar for cells cycled at varying current densities. We have also included EIS spectra for symmetric SS | LPSCI | SS cells for comparison (Supplementary Figure 9).

Meanwhile, the high frequency tail, which would correspond to the bulk impedance of the cell, can be expected to evolve as lithium metal plates onto the SS current collector. This can be attributed to a progressively improving contact between the SS foil and the SE. Schlenker et al. (ACS Appl. Mater. Interfaces 2020, 12, 17, 20012–20025) have previously reported such evolution as lithium metal plates onto an electrode. A brief discussion of these aspects has been included in the main text -

"The cells were allowed to stabilise over a period of 24-48 hours (see Figure S8a-b) prior to plating. A solid-state symmetric cell (SS|LPSCI|SS) was also setup for comparison (see Figures S9a-b) demonstrating the ion-blocking nature of the SS electrodes. As in the case of XPS spectra discussed earlier, the Nyquist plots also show marked differences in their

evolution for equivalent amounts of charge passed, depending on the current density applied (Figures 3a-d). The high frequency tail, which would correspond to the bulk impedance of the cell, can be expected to evolve as lithium metal plates onto the SS current collector. This can be attributed to a progressively improving contact between the SS foil and the SE. Schlenker et al. have previously reported such evolution as lithium metal plates onto an electrode.”

Supplementary Figure S9: (a) EIS spectra for symmetric SS | Li₆PS₅Cl | SS cells acquired over a 24 hour period demonstrating stabilisation of LPSCI with ion-blocking SS electrodes. (b) Open circuit potential stability measured across the same symmetric SS | LPSCI | SS cell over a 72 hour period.

Similarly, in the low frequency case, accepted to correspond to the SEI and other diffusion processes, the EIS spectra tails can be expected to evolve with continued plating. Further, we attribute the trends observed to a current density dependence of the SEI layer (Figure 3), wherein at higher current densities, the low frequency tail stabilises faster, facilitated by an Li₃P-rich SEI (Figure 2). This has been discussed in fair detail within the main text.

As correctly pointed out by the reviewer, a charge of $\sim 8.5 \mu\text{Ah}/\text{cm}^2$ would correspond to $\sim 40 \text{ nm}$ of lithium plated. However, most of the lithium is consumed in forming the SEI layer comprising decomposition products such as Li₂S, Li₃P and LiCl. It is difficult to estimate the critical lithium thickness to be plated for a stable SEI layer since this can depend on a variety of factors such as cell setup, stack pressure, temperature, and as we report here, current density during operation as well. There is still very little consensus on the thickness of the SEI layer formed on contact of LPSCI with Li metal. In recent work, Otto et al. investigated the Li-LPSCI interface (among other solid electrolytes) in-situ using ToF-SIMS and AFM [Otto2022]. Here, they report that the thickness of the Li₂S-rich SEI formed with LPSCI is $\sim 250 \pm 25 \text{ nm}$, for Li plated using an electron flood gun operated at $10 \mu\text{A}$ ($\sim 0.1 \text{ mA}/\text{cm}^2$) over 1h, inside the SIMS instrument. This thickness estimate is approximately 10 times greater than previous estimates of such SEI thicknesses [Wenzel2018].

REVIEWERS' COMMENTS

Reviewer #1 (Remarks to the Author):

The authors have answered all my open questions. Great manuscript.

Reviewer #2 (Remarks to the Author):

Dear authors,

Thank you for the detailed answer to my comments that bring clarity to your interesting work.

Therefore, here are some comments as reply to yours.

Regarding my question to the pressure inside the XPS chamber, I know that PHI system do not use ionic pump due to the presence of GCIB gun and there turbo is powerful enough to allow reasonable XPS analyses conditions. My meaning was, is if the pressure change while you are applying e-gun to the surface, signature of some degazing or surface evolution. If not, your answer is good enough.

The answer of the authors to both reviewers seems to bring some clarity to the paper.

Response to reviewer comments

We would like to thank the reviewers for their constructive and insightful comments. Our responses to questions and clarifications sought have been presented as below. Changes implemented to the original document are highlighted in yellow for the direct quote to the revised manuscript.

Reviewer #1:

The authors have answered all my open questions. Great manuscript.

We thank the reviewer for their comments on the manuscript.

Reviewer #2:

Dear authors,

Thank you for the detailed answer to my comments that bring clarity to your interesting work.

We thank the reviewer for their comments on the manuscript.

Therefore, here are some comments as reply to yours.

Regarding my question to the pressure inside the XPS chamber, I know that PHI system do not use ionic pump due to the presence of GCIB gun and there turbo is powerful enough to allow reasonable XPS analyses conditions. My meaning was, is if the pressure change while you are applying e-gun to the surface, signature of some degazing or surface evolution. If not, your answer is good enough.

Following the reviewer's clarification on the question of pressure inside the XPS chamber, we reviewed our XPS instrument conditions and checked for changes in chamber pressure upon repeatedly switching the electron gun between its on and off states. No significant changes in pressure were noticed which could possibly indicate lack of degassing or surface evolution as a consequence of this operation.

The answer of the authors to both reviewers seems to bring some clarity to the paper.